# A Direct Comparison between the Lateral Magnetophoretic Microseparator and AdnaTest for Isolating Prostate Circulating Tumor Cells

**DOI:** 10.3390/mi11090870

**Published:** 2020-09-19

**Authors:** Hyungseok Cho, Jae-Seung Chung, Ki-Ho Han

**Affiliations:** 1Department of Nanoscience and Engineering, Center for Nano Manufacturing, Inje University, Gimhae 50834, Korea; elshaddai88@naver.com; 2Department of Urology, Haeundae Paik Hospital, Inje University, Busan 48108, Korea

**Keywords:** circulating tumor cells, lateral magnetophoretic microseparator (CTC-µChip), AdnaTest, prostate cancer, microfluidics

## Abstract

Circulating tumor cells (CTCs) are important biomarkers for the diagnosis, prognosis, and treatment of cancer. However, because of their extreme rarity, a more precise technique for isolating CTCs is required to gain deeper insight into the characteristics of cancer. This study compares the performance of a lateral magnetophoretic microseparator (“CTC-μChip”), as a representative microfluidic device, and AdnaTest ProstateCancer (Qiagen), as a commercially available specialized method, for isolating CTCs from the blood of patients with prostate cancer. The enumeration and genetic analysis results of CTCs isolated via the two methods were compared under identical conditions. In the CTC enumeration experiment, the number of CTCs isolated by the CTC-μChip averaged 17.67 CTCs/mL, compared to 1.56 CTCs/mL by the AdnaTest. The number of contaminating white blood cells (WBCs) and the CTC purity with the CTC-μChip averaged 772.22 WBCs/mL and 3.91%, respectively, whereas those with the AdnaTest averaged 67.34 WBCs/mL and 1.98%, respectively. Through genetic analysis, using a cancer-specific gene panel (AR (androgen receptor), AR-V7 (A\androgen receptor variant-7), PSMA (prostate specific membrane antigen), KRT19 (cytokeratin-19), CD45 (PTPRC, Protein tyrosine phosphatase, receptor type, C)) with reverse transcription droplet digital PCR, three genes (AR, AR-V7, and PSMA) were more highly expressed in cells isolated by the CTC-μChip, while KRT19 and CD45 were similarly detected using both methods. Consequently, this study showed that the CTC-μChip can be used to isolate CTCs more reliably than AdnaTest ProstateCancer, as a specialized method for gene analysis of prostate CTCs, as well as more sensitively obtain cancer-associated gene expressions.

## 1. Introduction

Circulating tumor cells (CTCs) are an emerging biomarker in cancer biology and clinical research as they can reveal the landscape of cancer genes [1,2,3]. Furthermore, CTCs are acquired through blood-based liquid biopsy, enabling real-time monitoring of cancer owing to minimal invasiveness [4]. Macroscale techniques, such as magnetic-activated cell sorting (MACS) [5], CellSearch [6], CellCollector [7], ISET [8], and AdnaTest [9], are conventionally used to isolate CTCs owing to their simplicity, availability, and well-established procedures [10,11]. In particular, the CellSearch system—the first CTC isolation product to receive United States (US) Food and Drug Administration (FDA) approval—is the gold standard method [12,13,14], used for comparative analysis with other techniques for CTC isolation. With increasing emphasis on the performance of CTC isolation (e.g., recovery rate, purity, throughput, and viability), advanced microfluidic techniques, such as Parsortix [13], IsoFlux [14], and Vortex [15], have emerged owing to their advantages of high reproducibility, integrative capacity, low price, and ease of automation [10,16]. However, innovative techniques are still required to obtain more accurate CTC enumeration and pure CTCs owing to the extreme rarity of CTCs and the need of precise genetic analysis for personalized cancer treatment [17,18].

Among the macroscale techniques, the AdnaTest (Qiagen) is a representative method for detecting cancer-associated genes in CTCs isolated from various types of cancer, including breast [19], colon [20], ovarian [21], and prostate [22] cancer. Its ease of use and high sensitivity means that the AdnaTest is widely used in detecting gene expression profiling in CTCs and identifying potential prognostic and diagnostic biomarkers for cancer. Several studies [19,23,24] showed that the AdnaTest is more sensitive for gene detection than the CellSearch system. The AdnaTest was also used to detect the androgen receptor splice variant 7 (AR-V7), a rare genetic biomarker that may be associated with resistance to AR-targeting drugs in the treatment of metastatic castration-resistant prostate cancer (mCRPC) [25,26,27]. Furthermore, to predict the judgment of therapeutic regimens in breast cancer, multigene profiling in CTCs was conducted via the AdnaTest [28,29]. The AdnaTest can only be used to characterize, but not enumerate, the molecules in CTCs; thus, to obtain comprehensive cellular and molecular information of the CTCs, some studies [30,31] used the CellSearch system for enumeration and the AdnaTest to analyze their genes.

This study compares the performance of a lateral magnetophoretic microseparator (“CTC-µChip”) [32,33,34], as a representative microfluidic device, and the AdnaTest ProstateCancer, as a commercially available specialized method, for isolating CTCs from the blood of patients with prostate cancer. The enumeration and purity of CTCs isolated from patients with primary and metastatic prostate cancer using the CTC-µChip and AdnaTest were evaluated and compared. Cancer-associated genes in the CTCs isolated via the two methods were measured using reverse transcription droplet digital PCR (RT-ddPCR), thereby directly comparing their accuracy for CTC-based genetic analysis.

## 2. Materials and Methods

### 2.1. Experimental Design and Working Principle

To compare the performance of the CTC-µChip and the AdnaTest, the peripheral blood of patients with prostate cancer was divided into four samples, with equal volumes of 4–5 mL. Two were used in each of the two isolation methods, one in CTC enumeration and another in genetic analysis. The CTC-µChip is a specific microfluidic device for isolating CTCs on the basis of lateral magnetophoresis (Figure 1a) [34], as explained in Appendix A (Appendix A). Electroplated ferromagnetic wires (Ni_0.8_Fe_0.2_) were inlaid at the bottom of the microchannel and angled (*θ* = 5.7°) in the direction of flow. Blood samples were treated with a density gradient centrifugation to remove red blood cells (RBCs), followed by incubation with anti-EpCAM (Epithelial cell adhesion molecule) antibodies and immunomagnetic nanobeads (Human EpCAM Positive Selection Kit, STEMCELL Technologies, Vancouver, CA, USA), according to the manufacturer’s instructions. The prepared blood sample and phosphate-buffered saline (PBS) solution were injected into the CTC-µChip at a flow rate of 2 mL/h. The CTCs tagged with magnetic nanobeads then flowed along the ferromagnetic wires and were isolated at the CTC outlet.

The AdnaTest ProstateCancer is a highly sensitive method for characterizing gene expression in prostate CTCs and comprises an immunomagnetic CTC isolation system and a prostate cancer-associated gene analysis kit (Figure 1b). Blood samples were treated with immunomagnetic microbeads that can specifically bind to CTCs. The CTCs were then enriched from the treated blood samples using a magnet and subsequently washed. CTCs and contaminating white blood cells (WBCs), isolated by the CTC-µChip and the AdnaTest, were used to measure the number and purity of CTCs and the WBC depletion rate and to analyze cancer-associated gene expression by RT-ddPCR (Figure 1c).

### 2.2. Fabrication of the CTC-µChip

The CTC-µChip consists of a disposable microchannel superstrate and a reusable substrate, which can be assembled and disassembled via vacuum pressure (Figure 2a), as reported in a previous study [33]. The disposable microchannel superstrate contains a microchannel, two inlets, two outlets, and a vacuum trench for vacuum assembly (Figure 2b). The key fabrication process, which enables disposable use, is a technique that bonds a microstructured polydimethylsiloxane (PDMS) replica and a silicone-coated polyethylene terephthalate (PET) ultrathin film by oxygen plasma treatment to create the microchannel. The magnetic field generated at the top of the substrate can efficiently penetrate the PET film and control the cells passing through the microchannel due to the ultra-thin thickness of 12 µm. The top of the reusable substrate was inlaid with ferromagnetic wires (Figure 2c); these were placed on the centerline of two stacked neodymium–iron–boron permanent magnets. Then, the disposable microchannel superstrate was aligned to the substrate and assembled via a vacuum pressure of −50 kPa. Along with a uniform external magnetic field, the inlaid ferromagnetic wires were used to create a regularly repeated magnetic force pattern to manipulate the magnetized CTCs passing through the microchannel. The detailed fabrication process is explained in Appendix A (Appendix A).

### 2.3. Preparation of Blood Samples

Peripheral whole-blood samples were collected from 14 patients with primary and metastatic prostate cancer. Written informed consent was obtained from all patients, and the research design and protocol were approved by the Institutional Review Board of Haeundae Paik Hospital (HPIRB 2018-01-005-004). In total, 10 blood samples from patients with primary (*n* = 4) and metastatic (*n* = 6) prostate cancer were used for CTC enumeration, and a total of 14 blood samples were used for gene expression analysis. The information and clinical characteristics of the enrolled patients with prostate cancer are shown in Appendix A. The blood samples were drawn into 10 mL Vacutainer tubes, with an anticoagulant agent (BD Vacutainer, K_2_EDTA, 18.0 mg, Plymouth, UK), stored at 4 °C, and processed within 4 h.

To prepare the blood samples for the CTC-µChip, RBCs were first removed via density gradient centrifugation (700 g for 30 min) using a 1.119 g/mL Ficoll solution (Histopaque-1119, Sigma-Aldrich, St. Louis, MI, USA). After centrifugation, the buffy coat layer was transferred into 5 mL of ice-cold PBS with 0.2% bovine serum albumin (BSA) in a 50 mL conical tube to prevent cell death and aggregation. After washing twice, they were resuspended in 200 μL of ice-cold PBS with 0.2% BSA in a 1.5 mL microcentrifuge tube. The anti-EpCAM antibodies and immunomagnetic nanobeads were sequentially mixed with the resuspended sample and incubated for 60 min and 90 min, respectively, at 4 °C, according to the manufacturer’s instructions. The final sample was diluted with 800 μL of ice-cold PBS containing 0.2% BSA. Then, the prepared 1 mL sample was transferred to a sample injection syringe for CTC isolation using the CTC-µChip.

The blood sample preparation process for the Qiagen kit AdnaTest ProstateCancerPanel AR-V7 followed the manufacturer’s protocol. First, magnetic microbeads coated with anti-CTC antibodies were washed three times with 1 mL of PBS. After bead washing, 100 μL of beads were mixed with the untampered blood of a patient in a 15 mL collection tube. The tube was then continuously rotated at 5 rpm for 30 min at room temperature. The tube was placed in a magnet, and the supernatant was discarded to remove the WBCs and RBCs. Next, 5 mL of PBS was mixed with the remaining sample, and the tube was again placed in the magnet to isolate the CTC and magnetic bead complexes. This washing step was repeated three times. The samples isolated by the CTC-μChip and the AdnaTest were resuspended in 100 μL of cell-fixing reagent for CTC enumeration or lysed for RT-ddPCR to determine the gene expression levels of the selected five genes.

### 2.4. CTC Enumeration

For CTC enumeration, CTCs and WBCs isolated by the two methods were fixed using 100 µL of 4% paraformaldehyde for 10 min. The fixed cells were then incubated at 4 °C for 30 min with a nucleic acid fluorescent dye (DAPI, Vector Laboratories, Burlingame, CA, USA) to identify nuclei and anti-CD45 Alexa Fluor 647 antibodies (BioLegend, San Diego, CA, USA) to identify the WBCs. They were subsequently permeabilized for 10 min using 100 µL of 0.2% Triton X-100 (AMRESCO, West Chester, IL, USA) and incubated at 4 °C for 30 min with anti-pan-cytokeratin Alexa Fluor 488 antibodies (eBioscience, Waltham, MA, USA) to identify the CTCs. The fluorescently stained cells were then classified as either CTCs or WBCs using confocal microscopy imagery (LSM800, Carl Zeiss, Oberkochen, Germany).

### 2.5. Gene Expression Analysis Using RT-ddPCR

Along with CTC enumeration, the CTCs and WBCs isolated by the two methods were lysed to analyze the gene expression levels of the selected five genes that reflect reactivity to androgen hormones (AR and AR-V7), prostate cancer progression (PSMA), epithelial phenotype (KRT19), and leukocyte-specific marker (CD45). The detailed protocols for messenger RNA (mRNA) extraction and complementary DNA (cDNA) synthesis are explained in Appendix A. To increase the sensitivity of gene detection, multiplex PCR pre-amplification was performed before ddPCR, and its detailed protocol and pre-amplification primer sets are disclosed in Appendix A. The pre-amplified cDNA template was then diluted in a ratio of 1:10 to measure the gene expression levels of the five selected genes using ddPCR with PCR primer sets Appendix A. From seven sets of no-template control (NTC) tests, using ddPCR (Appendix A), the gene expression thresholds of the five genes were determined to be 0.5 copies/µL at AR, 0.79 copies/µL at AR-V7, 0.46 copies/µL at PSMA, 0.16 copies/µL at KRT19, and 0.27 copies/µL at CD45 (Appendix A). To reduce the false-positive rate, the gene expression thresholds were set to be 0.1 copies/µL higher than the maximum levels measured in the NTC tests. If a target gene was detected above the threshold value, it was considered as a positively detected sample. Gene expressions in the CTCs isolated from patients (*n* = 14) with primary (P1 to P5) and metastatic (P6 to P14) prostate cancer using the CTC-µChip and the AdnaTest were measured by ddPCR, as shown in Appendix A.

## 3. Results

### 3.1. Comparison of CTC Enumeration

CTCs and WBCs isolated by the two methods were identified by immunofluorescence staining. Then, CTCs and WBCs isolated by the CTC-µChip were clearly dyed and classified with anti-pan-cytokeratin Alexa 488 (green) and anti-CD45 Alexa 647 (red) antibodies, respectively, whereas cells isolated by the AdnaTest were surrounded by 4.8 µm diameter magnetic beads, making them difficult to distinguish (Figure 3). CTCs were detected in 14 out of 14 (100%) patients using the CTC-µChip and in nine out of 10 (90%) patients using the AdnaTest. The number of CTCs isolated by the CTC-µChip and the AdnaTest averaged 14.8 and 0.83 CTCs/mL, respectively, for primary cancer patients and 20.54 and 2.29 CTCs/mL, respectively, for metastatic cancer patients (Figure 4a). As expected, the results showed that the number of CTCs increased with the stage of prostate cancer. These results also indicated that the average number of CTCs isolated by the CTC-µChip was higher than that by the AdnaTest because the complexes of CTCs and magnetic beads in the AdnaTest easily attach to the tube wall and result in losses during transfer to a confocal dish.
(1)$$\mathrm{CTC}\;\mathrm{purity}\;(\%)\;=\;\frac{\mathrm{The}\;\mathrm{number}\;\mathrm{of}\;\mathrm{isolated}\;\mathrm{cytokeratin}-\mathrm{positive}\;\mathrm{cells}\;(\mathrm{CTCs})}{\mathrm{Total}\;\mathrm{number}\;\mathrm{of}\;\mathrm{cytokeratin}-\mathrm{positive}\;\mathrm{cells}\;(\mathrm{CTCs})\;\mathrm{and}\;\mathrm{CD}45-\mathrm{positive}\;\mathrm{cells}\;(\mathrm{WBCs})}\times 100.$$

Contaminating WBCs degrade the CTC purity, an important factor for precise genetic analysis. The numbers of contaminating WBCs by the CTC-µChip and AdnaTest were an average of 694.05 and 57.64 cells/mL, respectively, for primary cancer patients and 850.38 and 77.04 cells/mL, respectively, for metastatic cancer patients (Figure 4a). As in the case of CTCs, many WBCs adhered to the tube wall during the AdnaTest isolation, resulting in the loss of many WBCs. The purity of the CTCs could be calculated from the number of cytokeratin-positive cells (CTCs) and CD45-positive cells (WBCs), measured using a confocal microscope. In detail, the purity of each sample was calculated by the ratio of cytokeratin-positive cells (CTCs) to total number of isolated nucleated cells (cytokeratin-positive cells and CD45-positive cells, Equation (1)). Then, the average purity rate was calculated by adding the all case of purity rate and divide them by sum of sample cases (four samples for primary tumor stages and six samples for metastatic stages). Therefore, the purities of CTCs isolated by the CTC-µChip and the AdnaTest were 2.89% and 1.03%, respectively, for primary cancer patients and 4.92% and 2.92%, respectively, for metastatic cancer patients (Figure 4b). The enumeration of CTCs and WBCs isolated by the AdnaTest was inaccurate; many were lost during the harvesting process or were surrounded by magnetic microbeads, impeding identification.

The *t*-test was statistically analyzed for the isolated numbers of CTCs (*p* = 0.001, 95% confidence interval (CI), 18.2 ± 4.1 (mean ± SD) for CTC-µChip and 1.7 ± 0.98 (mean ± SD) for AdnaTest) and WBCs (*p* = 0.010, 95% CI, 787.8 ± 247.5 (mean ± SD) for CTC-µChip and 69.2 ± 26.4 (mean ± SD) for AdnaTest) to each method (Figure 4a). It resulted that the isolated numbers of CTCs and WBCs were significantly meaningful to each isolation method. The purity rates (*p* = 0.279, 95% CI, 4.1 ± 2.9% (mean ± SD) for CTC-µChip and 2.1 ± 1.8% (mean ± SD) for AdnaTest) were not significantly meaningful. Because of inaccuracy isolation during the enumeration process, the calculated numbers of CTCs and WBCs were not exact numbers of the AdnaTest that influenced the final calculation of purity rate, thereby drawing nonmeaningful statistical results.

### 3.2. Comparison of Gene Expression Analysis

In total, 14 patients with prostate cancer were enrolled for gene expression analysis in CTCs. The detection rate of the selected five genes was obtained from samples in which gene expression was over the threshold (Figure 5a). The detection rates of the AR gene were 85.71% (12/14 patients) and 28.57% (4/14 patients) for the CTC-µChip and the AdnaTest, respectively, whereas those of the AR-V7 gene were 14.29% (2/14 patients) and 7.14% (1/14 patients), respectively, and those of the PSMA gene were 57.14% (8/14 patients) and 42.86% (6/14 patients), respectively. The genes associated with reactivity to androgen hormones (AR and AR-V7) and prostate cancer progression (PSMA) were highly detected using the CTC-µChip. The genes KRT19 and CD45 were detected at the same rate in both isolation methods, i.e., 14.29% (2/14 patients) for KRT19 and 100% (14/14 patients) for CD45.

The average expression level for each of the four genes, except CD45, was higher with the CTC-µChip than with the AdnaTest, e.g., AR (1013.57 vs. 854.75 copies/µL, *p* = 0.09 with 95% CI), AR-V7 (39.15 vs. 3.5 copies/µL, *p* = 0.825 with 95% CI), PSMA (2100.22 vs. 1232.43 copies/µL, *p* = 0.454 with 95% CI), KRT19 (524.5 vs. 213.5 copies/µL, *p* = 0.468 with 95% CI), and CD45 (264.09 vs. 300.64 copies/µL, *p* = 0.920 with 95% CI), as shown in Figure 5b. Although AR gene expression was similar for the two methods, its detection rate was much higher with the CTC-µChip. The AR-V7 gene, associated with resistance to AR-targeting agents in patients with mCRPC, was only detected in metastatic cancer patients and displayed twice the detection rate, with an expression level that was 10 times higher for the CTC-µChip than for the AdnaTest. The PSMA and KRT19 genes exhibited twofold higher expression levels with the CTC-µChip. The expression levels of CD45 for the two isolation methods were the same, indicating that the number of contaminating WBCs was also the same for the two methods. The statistical *p*-value of gene expression level resulted not significantly different between both groups. It demonstrated that if a gene was detected by AdnaTest, the gene expression level also expressed a similar level to the CTC-µChip isolated sample. Although the gene expression level was not statistically meaningful, the detection rates of the three cancer-related genes (AR, AR-V7, and PSMA) were higher with the CTC-µChip, while two genes (KRT19 and CD45) had the same detection rate. Therefore, the gene expression analyses probably indicate that the number of CTCs was higher in the sample isolated by the CTC-µChip, while the number of contaminating WBCs was similar in both samples isolated by the two methods. 

Figure 6 shows the expression levels of the five selected genes in CTCs isolated from patients (*n* = 14) with primary and metastatic prostate cancer. It shows that most of the five genes were expressed at high levels in metastatic cancer patients. The AR gene was the most frequently detectable gene and more expressed through two stages isolated by CTC-µChip (Figure 6a). The AR-V7 gene was detected two cases (P7 and P14) at CTC-µChip and one case at AdnaTest (P13) in metastatic stages (Figure 6). In particular, the PSMA gene, a valuable biomarker for predicting the outcome in patients with prostate cancer, was detected in CTCs of many metastatic prostate cancer patients (57.14% (8/14 patients) at CTC-µChip and 42.86% (6/14 patients) at AdnaTest) (Figure 6c). This result is consistent with clinical studies [35,36], which reported the overexpression of PSMA in metastatic prostate patients. The KRT19 gene is a rarely detectable gene like AR-V7 and was especially highly expressed in patient 6 (P6) (Figure 6d). Interestingly, the concentration of serum PSA of patient 6 (P6) was 1921 ng/mL with a Gleason score of 10 (Appendix A) before the CTC test, demonstrating the high-risk patient situation. Therefore, the selected gene results showed the higher expression levels of AR and PSMA, as well as the KRT19 gene of patient 6 (P6), using both methods. More interestingly, none of the genes except for CD45 were detected in the AdnaTest in primary cancer patients, which means that CTCs might not have been isolated in primary cancer patients (Figure 6e). However, more genes were detected in metastatic cancer patients; this shows that more CTCs were isolated from metastatic cancer patients by both methods. 

## 4. Conclusions

The CTC-µChip, as a representative microfluidic device, and the AdnaTest ProstateCancer, as a commercially available method, for isolating CTCs from patients with prostate cancer, were directly compared with the performance of CTC enumeration and CTC-based genetic analysis. The number of CTCs isolated by the CTC-µChip averaged 17.67 CTCs/mL, higher than 1.56 CTCs/mL by the AdnaTest. The number of contaminating WBCs and the CTC purity averaged 772.22 WBCs/mL and 3.91%, respectively, on the CTC-µChip, and 67.34 WBCs/mL and 1.98%, respectively, on the AdnaTest. In the AdnaTest, the number of isolated cells (CTCs and WBCs) was relatively small owing to the complexes of cells and magnetic beads attached to the tube wall for the isolation process, resulting in losses during transfer to a confocal dish. The results revealed that the AdnaTest is not suitable for CTC enumeration.

The expression level and detection rate of the five selected genes in CTCs isolated by the two methods were measured using RT-ddPCR, from which the data of the CTC-µChip were determined to be much higher than those of the AdnaTest. As expected, the expression of cancer-related genes in CTCs increased as the stage of prostate cancer advanced and, interestingly, no genes were detected with the AdnaTest in patients with primary prostate cancer, except for CD45. Although the AdnaTest is a specialized method for analyzing prostate cancer-associated gene expression in CTCs, the CTC-µChip was used to detect much higher CTC-based gene expression in all stages of prostate cancer. Consequently, this study clearly demonstrates that the CTC-µChip, as a microfluidic device, can be applied for both precise CTC enumeration and CTC-based genetic analysis.

## Figures and Tables

**Figure 1 micromachines-11-00870-f001:**
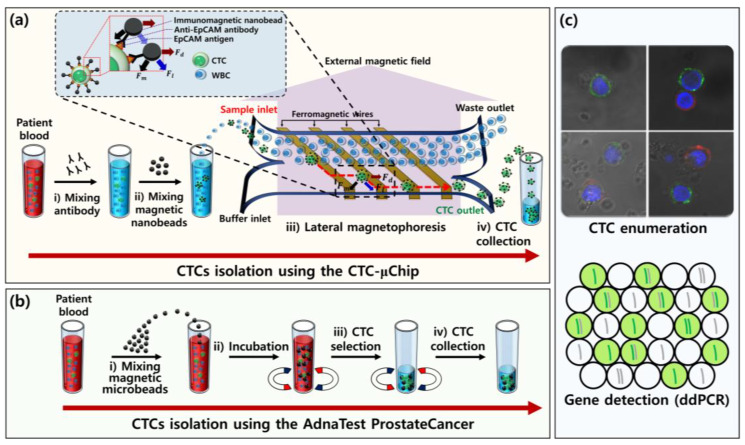
Illustration of the experimental workflows to isolate circulating tumor cells (CTCs) with the CTC-µChip and the AdnaTest ProstateCancer. (**a**) For the CTC-µChip, blood samples were treated with a density gradient centrifugation to remove the red blood cells (RBCs), followed by incubation with (i) anti-EpCAM antibodies and (ii) immunomagnetic nanobeads, in sequence. Then, (iii) the prepared blood sample and phosphate-buffered saline (PBS) were injected into the CTC-µChip, and (iv) the CTCs flowed along the ferromagnetic wires and were isolated at the CTC outlet. (**b**) For the AdnaTest ProstateCancer, (i) whole blood was mixed and (ii) incubated with immunomagnetic microbeads. (iii) The treated blood samples were placed in a magnet to collect the CTCs, and (iv) the supernatant was discarded to enrich the CTCs. (**c**) Downstream analyses were performed to enumerate the CTCs and characterize their gene expression using RT-ddPCR.

**Figure 2 micromachines-11-00870-f002:**
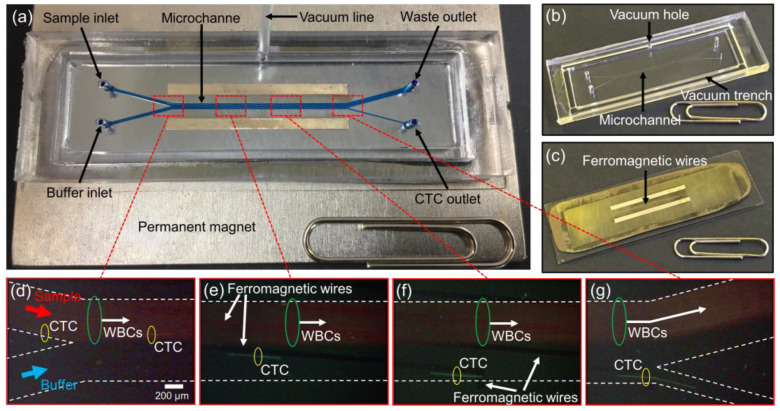
Photographs of (**a**) the CTC-µChip, consisting of (**b**) the disposable microchannel superstrate and (**c**) the reusable substrate, assembled via vacuum pressure. Enlarged views of (**d**) the sample and buffer inlets, (**e**) the early and (**f**) late points of the microchannel, and (**g**) the waste and CTC outlets.

**Figure 3 micromachines-11-00870-f003:**
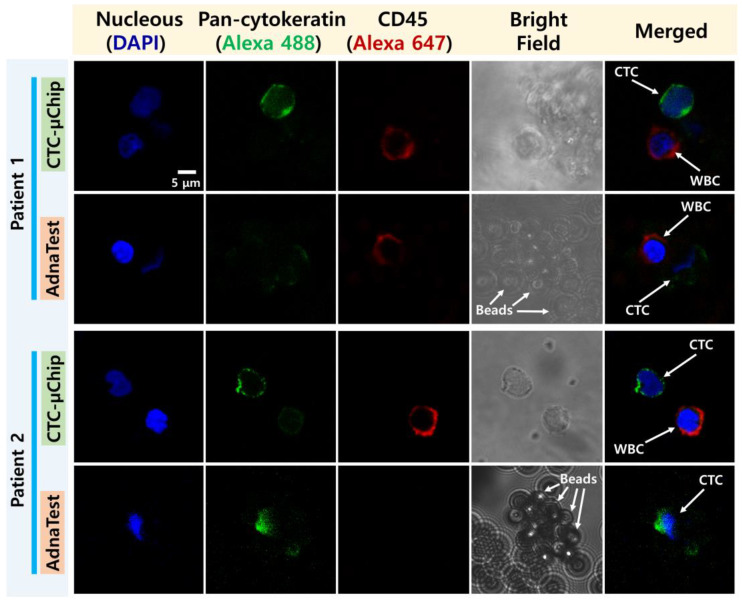
Immunofluorescence confocal microscopy images of prostate CTCs and white blood cells (WBCs) isolated by the CTC-µChip and AdnaTest.

**Figure 4 micromachines-11-00870-f004:**
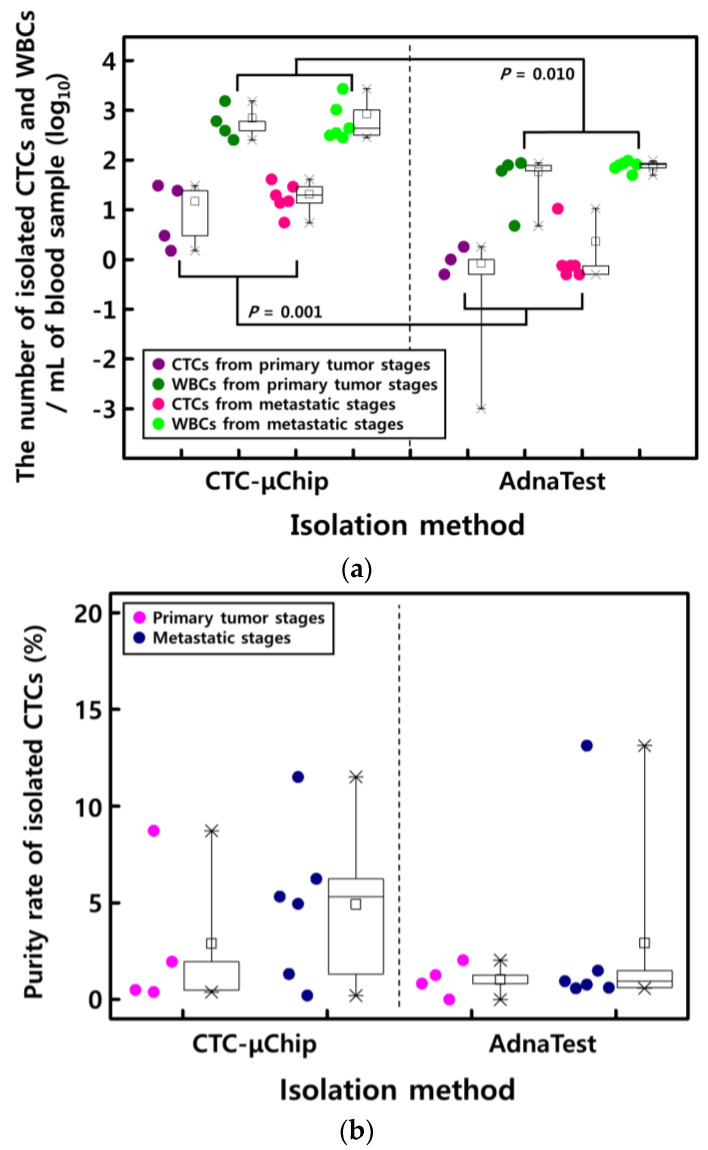
(**a**) The number of isolated CTCs and WBCs per milliliter of blood, with the CTC-µChip and AdnaTest. (**b**) The purity of CTCs in patients with primary and metastatic prostate cancer.

**Figure 5 micromachines-11-00870-f005:**
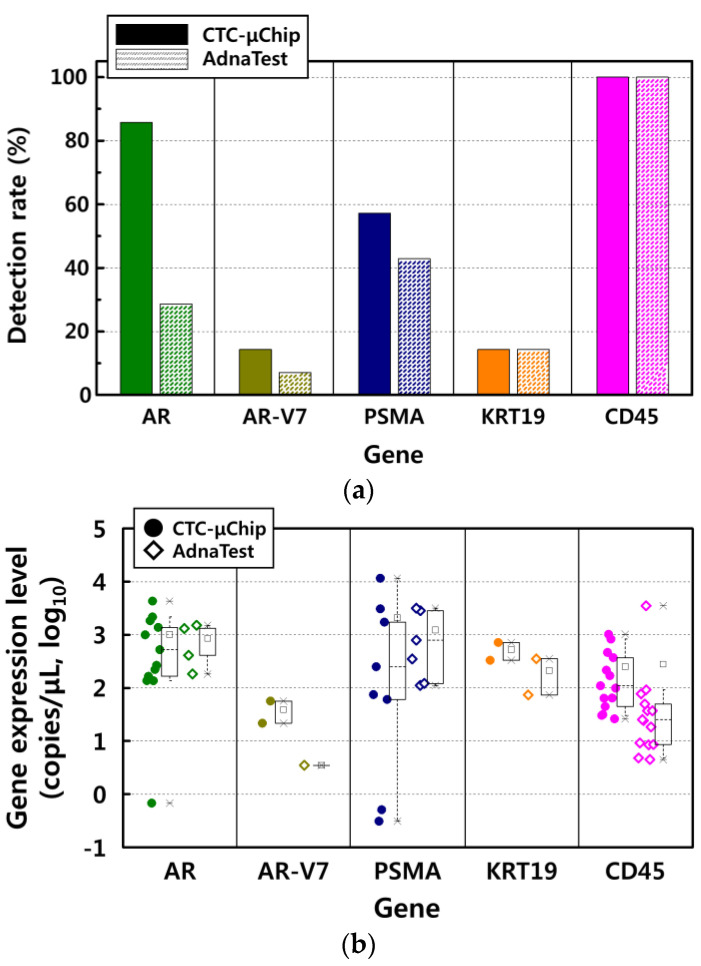
(**a**) The detection rate and (**b**) expression levels (copies/µL) of the selected five genes of androgen receptor (AR), androgen receptor splice variant 7 (AR-V7), prostate-specific membrane antigen (PSMA), cytokeratin 19 (KRT19), and protein tyrosine phosphatase receptor type C (CD45). The *p*-value of gene expression was statistically analyzed for the two isolation methods, giving results of 0.09 for AR, 0.825 for AR-V7, 0.454 for PSMA, 0.468 for KRT19, and 0.920 for CD45 with 95% confidence interval (CI).

**Figure 6 micromachines-11-00870-f006:**
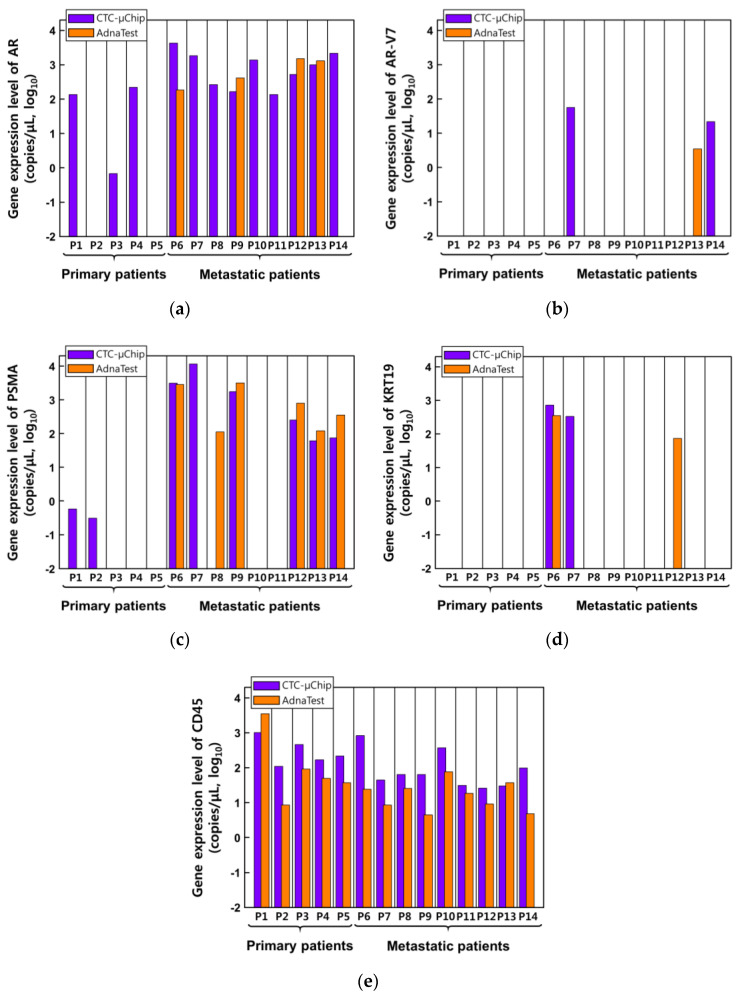
The expression levels of the selected five genes (**a**) AR, (**b**) AR-V7, (**c**) PSMA, (**d**) KRT19, and (**e**) CD45) in CTCs isolated from patients with prostate cancer. Primary cancer patients are marked P1 to P5 (*n* = 5), and metastatic cancer patients are marked P6 to P14 (*n* = 9).

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
