# Peer review of "A Direct Comparison between the Lateral Magnetophoretic Microseparator and AdnaTest for Isolating Prostate Circulating Tumor Cells"

_micromachines, 2020, doi:10.3390/mi11090870_

Round 1
Reviewer 1 Report
The authors offer an insightful study where they evaluate magnetic lateral separation of CTCs with respect to the existing AdnaTest ProstateCancer. The study underscores magnetic lateral separation in a microfluidic system as a platform that outperforms currently existing technologies in capturing CTCs. The study is of importance to the biomedical engineering and biotechnology fields as well as generally to the cancer research field and specifically for the community of translational diagnostic devices.
Although the authors use preexisting technology (labeling/nanoparticle tagging) in combination with previously described platform designs, this study counts as a very important contribution to the field. The authors should comment on why this particular preprocessing works best among other known microfluidic technologies in literature. For example, The authors have previously describe several of such designs for the separation of WBCs and RBCs using intrinsic and differential magnetic properties of WBCs and RBCs (e.g. Applied physics letters 2008). Other platforms that I am familiar with and worked on include the use of Gd-DTPA magnetic contrasting agents to emphasize the magnetic susceptibility difference between cells types. This has generally been used in cell magnetophoretic separation but also to create 3D cellular constructs. Can any of these other technologies work just as well as tagging? The likely answer is perhaps not due to lower specificity and higher contamination, which may render detection a hard task especially for rare CTCs. Perhaps the authors can quickly in a line or two discuss the drawbacks of other microfluidic platforms and why this one is the obvious choice for CTC detection.
The authors should also explore briefly (in discussion form) as to why the CTC-microCHIP outperforms the AdnaTest. Is it perhaps the high gradient magnetic fields (HGMFs) created by the deposited ferromagnetic materials? Can this setup create much larger forces (due to the nature of HGMFs created by the induced magnetization) compared to the setup of the AdnaTest and therefore “pull” more CTCs to be detected. In other words, is the purity of detection a function of the magnetic field gradient. If yes, then having HGMFs will of course create larger forces though larger values of grad|B|2 and thus will allow for better enrichment. However, HGMFs at the cell scale demands miniaturization which leads to the adoption of a microfluidic system in the form of the CTC-microCHIP, and that may be why the described platform outperforms the AdnaTest.
Overall the study is well founded. There are however minor adjustments that the authors should consider addressing some of them to render this study more impactful.
Figure 1a and S1:
It is not immediately clear that the CTCs have been labeled in the channel. To help the reader form a streamlined process it may be beneficial to add a small cartoon call-out where the CTC that’s above the magnetic wire shows its labeled nanobead and that the source of the guiding force is from the magnetic nanobeads.
Figure 4
Using the asterisks symbol for denoting local and metastatic stages/patients may be confusing since often these symbols are reserved for indicating significance.
The authors should indicate whether this is a log10 graph since often in biological papers there is the use of log2 for example in the log fold change of a gene.
The authors should consider conducting some statistics to validate their claims on this important graph. For example, a t-test with 95% CI and appropriate corrections for normality and SDs. The authors should then report these values in the text as a “punchline” for their findings. The authors can utilize a program such as GraphPad or R to conduct statistical significance tests.
Line 191-201
Regarding the CTC purity. It is no very clear what is the equation used here. Is it simply the ratio CTCs/(CTCs+WBCs)? If that is the case then would it be correct to estimate the purity for example of the metastatic patients by using the average described CTC numbers for CTC-microCHIP (20.54 CTCs/mL in line 186) and the corresponding contaminating WBCs (850.38 WBC/mL in line 193)? With these numbers the fraction of CTCs is ~ 2.36%. The same analysis done for the AdnaTest for the metastatic patients yields a CTC fraction of 2.89%, i.e. slightly purer but likely not significant than the CTC-microCHIP. These overly simplified calculations reflect differently than what the authors have indicated in the text. Can the authors comment on this analysis as it is very likely that I missed something in the purity calculation. If this is incorrect then it may be possible for the authors to streamline this part of the manuscript so as to not cause any confusion. This can likely be done by describing exactly how the purity calculation is done.
The authors should investigate the statistics on the purity between the two platforms.
Figure 5a:
The authors should clarify how the detection rate was derived, i.e. what does 100%, 50% mean in this context. The authors indicate (line 225) that the number of WBC contaminating cells are the same for both methods of detection based on the CD45 signal. However, in the Figure 4, it was shown that the WBCs detected using the AdnaTest was ~10 times lower than the CTC-microCHIP.
Similarly, Line 226-228:
From the gene expression profile, it seems obvious that the CTC markers are indeed higher for the CTC-microCHIP platform. This agrees with the previous protein level observations in Figure 4, i.e. that there are more CTCs detected by this method than the AdnaTest. However, gene expression and protein detection observations for WBCs do not agree. The authors should discuss as to the possible sources of this discrepancy. For example, that mRNA levels do not necessarily always have to exactly match protein observations?
Figure 5b:
The authors should again use statistics here to describe sample groups where possible.
Line 236:
It is more valuable here specifically to give hard numbers instead of using “many”. For example, “…was detected in CTCs of X out of Y metastatic prostate cancer patients”. This gives a better feel for the technology for the reader. This number should be reported for both platforms. Although this can be derived from the graph in Figure 6, it is better to also write down that number.
Figure 6:
The author should consider having a small graph (pie chart / doughnut chart or another bar graph) to show the total patients where a gene was detected for both platforms.
Given that most CTC genes were not detected in the local cancer patients (only AR and PSMA in some) for the CTC-microCHIP, should this be a positive hit? This leads to the questions of what is the minimum criteria for a positive hit at the mRNA level? It seems that AR and PSMA are the most frequently upregulated genes. Should they be present together for a “confirmed” positive? This should be addressed since some patients have only 1 (CTC) gene detected. The reviewer does not know any further, and perhaps one gene is sufficient for a “confirmed” positive, however, this can be emphasized in the text.
General comment:
-It may be more traditional to use the words “primary tumor patients” instead of the word “Local”. Then the two patient groups will be a primary tumor and metastatic stages.
-For all the results, have the authors conducted any controls? e.g. any peripheral blood samples from none cancer patients where the tests for the CTC-microCHIP and AdnaTest both show 0 detection of CTC on the protein and mRNA levels (or even just for the protein through IHC)? I believe this controls will be important to validate the technology further.
Author Response
1. Comment: Figure 1a and S1:
It is not immediately clear that the CTCs have been labeled in the channel.
To help the reader form a streamlined process it may be beneficial to add a small cartoon call-out where the CTC that’s above the magnetic wire shows its labeled nanobead and that the source of the guiding force is from the magnetic nanobeads.
Answer:
Figures 1(a) and S1 were revised by inserting an enlarged cartoon of CTC labeled with immunomagnetic nanobeads. The cartoon was used in Figs. 1(a) and S1 to clarify that CTCs were first labeled with the magnetic beads and then isolated in the CTC-µChip. It was also explained in the caption to Fig. 1 (lines 93 to 95) and Section 2.3 (lines 138 to 139) as highlighted in yellow.
2. Comment: (2-1) Figure 4
Using the asterisks symbol for denoting local and metastatic stages/patients may be confusing since often these symbols are reserved for indicating significance.
(2-2) The authors should indicate whether this is a log10 graph since often in biological papers there is the use of log2 for example in the log fold change of a gene.
(2-3) The authors should consider conducting some statistics to validate their claims on this important graph. For example, a t-test with 95% CI and appropriate corrections for normality and SDs. The authors should then report these values in the text as a “punchline” for their findings. The authors can utilize a program such as GraphPad or R to conduct statistical significance tests.
Answer:
(2-1) In Fig 4, the asterisks symbols were removed and replaced by colors.
(2-2) The scale of y-axis in Figs. 4(a), 5(b), and 6 was indicated as log10.
(2-3) The numbers of isolated CTCs and WBCs were statistically analyzed by the t-test using SPSS software. The analyzed statistical p value and SDs were discussed on lines 214 to 222 and highlighted in yellow.
3. Comment: (3-1) Line 191-201
Regarding the CTC purity. It is no very clear what is the equation used here.
Is it simply the ratio CTCs/(CTCs+WBCs)? If that is the case then would it be correct to estimate the purity for example of the metastatic patients by using the average described CTC numbers for CTC-microCHIP (20.54 CTCs/mL in line 186) and the corresponding contaminating WBCs (850.38 WBC/mL in line 193)? With these numbers the fraction of CTCs is ~ 2.36%. The same analysis done for the AdnaTest for the metastatic patients yields a CTC fraction of 2.89%, i.e. slightly purer but likely not significant than the CTC-microCHIP. These overly simplified calculations reflect differently than what the authors have indicated in the text. Can the authors comment on this analysis as it is very likely that I missed something in the purity calculation. If this is incorrect then it may be possible for the authors to streamline this part of the manuscript so as to not cause any confusion. This can likely be done by describing exactly how the purity calculation is done.
(3-2) The authors should investigate the statistics on the purity between the two platforms.
Answer:
(3-1) To clarify how the purity of isolated CTCs is calculated, the purity equation was added to lines 194 to 198 as highlighted in yellow. The CTC purity calculated by the average number of CTCs and WBCs for the entire patient may differ from the result calculated by the CTC purity for each patient. Raw data on the number of CTCs and WBCs isolated from each patient and their CTC purity were disclosed in Table S1.
(3-2) The purity t-test of isolated CTCs was analyzed and discussed on lines 218 to 222 as highlighted in yellow.
4. Comment: (4-1) Figure 5a:
The authors should clarify how the detection rate was derived, i.e. what does 100%, 50% mean in this context.
(4-2) The authors indicate (line 225) that the number of WBC contaminating cells are the same for both methods of detection based on the CD45 signal. However, in the Figure 4, it was shown that the WBCs detected using the AdnaTest was ~10 times lower than the CTC-microCHIP.
Similarly, Line 226-228:
From the gene expression profile, it seems obvious that the CTC markers are indeed higher for the CTC-microCHIP platform. This agrees with the previous protein level observations in Figure 4, i.e. that there are more CTCs detected by this method than the AdnaTest. However, gene expression and protein detection observations for WBCs do not agree. The authors should discuss as to the possible sources of this discrepancy. For example, that mRNA levels do not necessarily always have to exactly match protein observations?
Answer:
(4-1) The detection rate was derived as a ratio of positively detected samples to the entire samples (14 samples) at each gene. For example, in case of the CTC-µChip, as the AR gene was detected in 12 of the 14 samples, its detection rate could be calculated at 85.71% (12/14). The number of positively detected samples for each gene was added to Section 3.2 (lines 234 to 240) and highlighted in yellow.
(4-2) After CTC isolation using the AdnaTest, many of the isolated cells were lost in the process of being transferred to a confocal dish for enumeration. It was described on lines 192-193, 202-203, and 301-304 and highlighted in yellow.
On the other hand, all isolated CTCs and WBCs could be used without loss in genetic analysis. For this reason, it was estimated that the number of WBCs used in genetic analysis would be the same in both isolation methods based on the similar expression levels of CD45, as described in lines 250 to 252 highlighted in yellow.
5. Comment: Figure 5b:
The authors should again use statistics here to describe sample groups where possible.
Answer:
The statistical p value of gene expression was analyzed and described in lines 242 to 245 and 252 to 257 as highlighted in yellow. In addition, it was indicated in the caption of Fig. 5.
6. Comment: Line 236:
It is more valuable here specifically to give hard numbers instead of using “many”. For example, “…was detected in CTCs of X out of Y metastatic prostate cancer patients”. This gives a better feel for the technology for the reader. This number should be reported for both platforms. Although this can be derived from the graph in Figure 6, it is better to also write down that number.
Answer:
The sentence was revised as “… was detected in CTCs of many metastatic prostate cancer patients [57.14% (8/14 patients) at CTC-µChip and 42.86% (6/14 patients) at AdnaTest] (Figure 6(c)).” in line 276 and highlighted in yellow.
In addition, the detailed detection rates for each gene were indicated at Section 3.2 (lines 234 to 240) and highlighted with yellow.
7. Comment: (7-1) Figure 6:
The author should consider having a small graph (pie chart / doughnut chart or another bar graph) to show the total patients where a gene was detected for both platforms.
(7-2) Given that most CTC genes were not detected in the local cancer patients (only AR and PSMA in some) for the CTC-microCHIP, should this be a positive hit? This leads to the questions of what is the minimum criteria for a positive hit at the mRNA level?
(7-3) It seems that AR and PSMA are the most frequently upregulated genes. Should they be present together for a “confirmed” positive? This should be addressed since some patients have only 1 (CTC) gene detected. The reviewer does not know any further, and perhaps one gene is sufficient for a “confirmed” positive, however, this can be emphasized in the text.
Answer:
(7-1) Figure 6 was revised to show five gene expression levels individually. In addition, it was discussed on lines 271 to 284, highlighted in yellow.
(7-2) The criteria for the positive hit of mRNA gene expression level (the gene expression threshold) was determined using seven NTC samples and demonstrated at lines 171 to 174 of Section 2.5 as highlighted in yellow. The evaluated threshold results were disclosed in Figs. S3 and S4 as supplementary materials.
If a target gene is detected above the threshold value, it was considered as a positively detected sample. It was explained in Section 2.5 on lines 176 to 177 as highlighted in yellow.
(7-3) Present data is insufficient to show which genes or combinations can be used as predictive biomarkers for “confirmed” positive, mentioned by the reviewer. This requires more genetic analysis based on CTCs, and the aim of this paper is to show which CTC isolation method (CTC-μChip or AdnaTest) can isolate CTCs more efficiently and use them for genetic analysis.
8. Comment: (8-1) General comment:
It may be more traditional to use the words “primary tumor patients” instead of the word “Local”. Then the two patient groups will be a primary tumor and metastatic stages.
(8-2) For all the results, have the authors conducted any controls? e.g. any peripheral blood samples from none cancer patients where the tests for the CTC-microCHIP and AdnaTest both show 0 detection of CTC on the protein and mRNA levels (or even just for the protein through IHC)? I believe this controls will be important to validate the technology further.
Answer:
(8-1) The words “local” and “localized” were revised to “primary”. The word “primary” was highlighted with yellow in main and supplementary manuscripts.
(8-2) The aim of this study is to compare the performance of the CTC-μChip and the AdnaTest and therefore, blood samples from healthy donors were not necessarily used as controls. However, in our previous study (Cho et al., Lab on a Chip 2017, 17, p.4113) and laboratory studies (unpublished), healthy blood samples were used with the CTC-μChip and no CTCs were detected.

Reviewer 2 Report
The authors compared the CTC-μChip with the AdnaTest for isolating CTCs from prostate cancer patients. Here are a few minor suggestions for the publication.
- At line 77, please describe the product information of anti-EpCAM antibodies.
- At line 103-117, is this CTC-μChip a new device? Or the same device you have been used? If the device is new or modified, please explain any modification or improvement of the device.
- At line 133-134, the centrifugation process may lose some CTCs. Do you have any reference this method is okay?
- At line 135, why did you use the ice-cold PBS, not room temperature PBS?
Author Response
- Comment: At line 77, please describe the product information of anti-EpCAM antibodies.
Answer: The product information (Human EpCAM Positive Selection Kit, STEMCELL Technologies) was disclosed on line 78 and highlighted in yellow.
- Comment: At line 103-117, is this CTC-μChip a new device? Or the same device you have been used? If the device is new or modified, please explain any modification or improvement of the device.
Answer: The original CTC-μChip was developed in 2013 (Kim et al., Analytical chemistry 2013, 85, p.2779) as the first version of the lateral magnetophoretic microseparator. The second version of CTC-μChip was developed in 2017 (Cho et al., Lab on a Chip 2017, 17, p.4113) as a disposable format for medical applications. The disposable CTC-μChip was used in this study.
- Comment: At line 133-134, the centrifugation process may lose some CTCs. Do you have any
reference this method is okay?
Answer: The sample preparation has been also conducted in our previous studies (Cho et al., Lab on a Chip 2017, 17, p.4113 and Kang et al., Micromachines 2019, 10, p.386). The studies showed that CTCs spiked into whole peripheral blood can be retrieved by approximately 95 to 98% through the entire process, including sample preparation and CTC isolation. Thus, CTC loss in the centrifugation process can be predicted to be 0 to 2% and therefore, we believe that the centrifugation process does not affect the performance of CTC isolation
- Comment: At line 135, why did you use the ice-cold PBS, not room temperature PBS
Answer: To prevent cell death and aggregation, ice-cold PBS was used at whole sample preparation. It was explained on line 136 and highlighted in yellow.
